# Biomimetic Red Blood Cell Membrane-Mediated Nanodrugs Loading Ursolic Acid for Targeting NSCLC Therapy

**DOI:** 10.3390/cancers14184520

**Published:** 2022-09-18

**Authors:** Ting Wu, Dan Yan, Wenjun Hou, Hui Jiang, Min Wu, Yanling Wang, Gang Chen, Chunming Tang, Yijun Wang, Huae Xu

**Affiliations:** 1Jiangsu Key Laboratory of Molecular and Translational Cancer Research, Jiangsu Institute of Cancer Research, Affiliated Cancer Hospital of Nanjing Medical University, Jiangsu Cancer Hospital, Nanjing 210009, China; 2Department of Pharmaceutics, School of Pharmacy, Nanjing Medical University, Nanjing 211116, China; 3Department of Dermatology, Drum Tower Hospital of Medical School, Nanjing University, Nanjing 211116, China; 4Department of Gastrointestinal Surgery, The Affiliated Jiangning Hospital of Nanjing Medical University, Nanjing 211199, China; 5Department of Pharmacy, The Second Affiliated Hospital of Nanjing Medical University, Nanjing 211116, China

**Keywords:** non-small cell lung cancer, ursolic acid, biomimetic nanocarriers, red blood cell membranes, anticancer therapeutics, apoptosis, autophagy

## Abstract

**Simple Summary:**

Lung cancer is the second most common cancer after breast cancer. Non-small-cell lung cancer, which represents more than 85% of all lung cancer subtypes, is known for its tumor progression and metastasis, resulting in poor clinical outcomes. Conventional therapies for NSCLC, such as surgery, chemotherapy, and radiotherapy, always fail due to therapeutic resistance. In recent years, ursolic acid (UA), a natural pentacyclic triterpenoid compound, has been shown to be a promising antitumor drug by regulating multiple signaling pathways in cancers. Unfortunately, the poor water solubility, low bioavailability, and systemic toxicity of UA limit its clinical application. In this study, a biomimetic red blood cell membrane nanocarrier was developed to deliver UA to targeted tumor sites efficiently, and it inhibited tumor growth by inducing the apoptosis and autophagy of cancer cells both in vitro and in vivo.

**Abstract:**

As one of the most common cancers worldwide, non-small-cell lung cancer (NSCLC) treatment always fails owing to the tumor microenvironment and resistance. UA, a traditional Chinese medicine, was reported to have antitumor potential in tumor models in vitro and in vivo, but showed impressive results in its potential application for poor water solubility. In this study, a novel biomimetic drug-delivery system based on UA-loaded nanoparticles (UaNPs) with a red blood cell membrane (RBCM) coating was developed. The RBCM-coated UANPs (UMNPs) exhibited improved water solubility, high stability, good biosafety, and efficient tumor accumulation. Importantly, the excellent antitumor efficiency of the UMNPs was confirmed both in vitro and in vivo in cancer models. In addition, we further investigated the antitumor mechanism of UMNPs. The results of Western blotting showed that UMNPs exerted an anticancer effect by inducing the apoptosis and autophagy of NSCLC cells, which makes it superior to free UA. In addition, body weight monitoring, hematoxylin and eosin (HE) analysis, and immunohistochemical (IHC) analysis showed no significant difference between UMNPs and the control group, indicating the safety of UMNPs. Altogether, the preparation of biomimetic UMNPs provides a promising strategy to improve outcomes in NSCLC.

## 1. Introduction

Worldwide, lung cancer is one of the most common cancers and the leading cause of cancer-related death, with 1.76 million deaths every year [1,2,3,4]. In particular, non-small-cell lung cancer (NSCLC), accounting for more than 85% of all lung cancer cases, presents a much poorer prognosis for patients [5,6]. Despite the considerable amount of standard first-line treatment methods such as surgery, chemotherapy, radiotherapy, targeted therapy, and immunotherapy being applied in clinics [7,8], many challenges remain, including distant metastasis after surgery, chemotherapy resistance, low radiosensitivity of cancer cells, lack of drug-targetable receptors, and an immunosuppressive tumor microenvironment (ITM) [9,10,11]. Additionally, the adverse effects of these treatments limit their application. Thus, developing novel treatment options to enhance the therapeutic effect for NSCLC is of great significance. 

In recent years, traditional Chinese medicines have received increased attention for the treatment of cancer [12,13]. One such traditional medicine is ursolic acid (UA), a kind of pentacyclic triterpenoid, which is widely distributed in certain medicinal herbs [14,15]. UA exhibits multiple biological activities and presents great potential for treating various diseases [16,17,18,19]. The antitumor activity and underlying antitumor mechanisms of UA, such as the induction of apoptosis and autophagic flux in several types of tumors, were verified in vitro and in vivo [17,20,21,22]. Unfortunately, its poor water solubility, low bioavailability, and systemic toxicity limit its clinical application [23,24]. Therefore, nanotechnology is being explored to design novel UA formulations with enhanced water solubility, better bioavailability, and biosafety to expand the clinical benefits [25,26]. 

Biodegradable polymeric nanoparticles, such as poly(ethylene glycol)-b-poly(epsilon-caprolactone) (PEG-PCL) and poly(lactic-co-glycolic acid) (PLGA), can improve the aforementioned problems and exhibit outstanding drug-loading capacity for controlling drug release [27], and were approved by the Food and Drug Administration (FDA) for drug delivery in various biomedical applications [28]. However, these PEGylated formulations may lead to the production of antiPEG IgG in vivo, following the immune response to and elimination of antiPEG by the reticuloendothelial system (RES) [29,30]. These PEGylated nanoparticles accelerate the blood clearance of PEG-mediated drugs, thereby reducing drug accumulation in tumor sites, leading to the failure of antitumor treatment. Thus, research into advanced nano-based drug delivery systems (NDDSs) is required to offer more opportunities for UA delivery.

Important advancements in biomimetic cell membrane-based nanocarriers to increase drug accumulation in tumor tissues have been achieved in recent decades [31]. Among them, red blood cell membranes (RBCMs), which are derived from red blood cells, the most numerous blood cells in the circulation, were shown to have excellent long-term circulatory efficiency for CD47 on the surface of membranes [32]. CD47 represents a “don’t eat me” signal and effectively avoids the clearance of immunity [33]. RBCM-coated nanoparticles can prolong the circulation of drugs in the blood, and then these nanosize drug-delivering carriers further target and enhance drug accumulation in tumor sites via the enhanced permeation and retention (EPR) effect. Consequently, biomimetic RBCM-coated nanocarriers further open possibilities for expanding the benefits of cancer treatment.

Here, we designed a biomimetic RBCM-based nanoplatform loaded with UA for the treatment of NSCLC. The nanoplatform UMNPs had a core–shell-like structure. The core consisted of UA-loaded polymeric nanoparticles constructed from PEG-PCL, coated with RBCMs acting as the shell. We then explored the antitumor effect of UMNPs in vitro and in vivo.

## 2. Materials and Methods

### 2.1. Materials

Ursolic acid (≥99.0%) and coumarin-6 (≥98.0%) were purchased from Macline Biochemical Technology Co., Ltd. (Shanghai, China). Poloxamer 188 PRO was purchased from Sigma-Aldrich (Shanghai, China). 1,1′-Dioctadecyl-3,3,3′,3′-tetramethylindocarbocyanine perchlorate (Dil), BCA protein assay kit, Coomassie blue staining solution, and 3-(4,5-dimethyl-2-thiazolyl)-2,5-diphenyl-2-H-tetrazolium bromide (MTT) were all purchased from Beyotime Biotechnology Co., Ltd. (Nantong, China). Crystalline violet staining solution, 4% paraformaldehyde, APC annexin V apoptosis detection kit, and reactive oxygen species assay kit were purchased from Yeasen (Shanghai, China).

### 2.2. Animals and Cell Lines

Male BALB/c nude mice 4–6 weeks old were obtained from the Animal Core Facility of Nanjing Medical University. All animal experimental procedures were approved by the Nanjing Medical University Ethics Committee for Animal Laboratory Research and followed the guidelines of ethical regulations for institutional animal care and use at Nanjing Medical University (approval number: 2104050).

Human A549 cells and NCI-H1975 cells and mouse macrophage RAW264.7 cells were obtained from the Institute of Biochemistry and Cell Biology, Shanghai Institute of Biological Sciences, Chinese Academy of Sciences (Shanghai, China), and cultured in RPMI-1640 and DMEM medium supplemented with 10% FBS in an incubator containing 5% CO_2_ at 37 °C.

### 2.3. Preparation of UA Nanoparticles (UaNPs)

UaNPs were synthesized by the nanoprecipitation method. Briefly, 4 mg UA and 20 mg PCL-PEG were dissolved in acetone as the organic phase. Ultrapure water was prepared as the aqueous phase. Under magnetic stirring, a 1 mL organic phase was slowly added to the 10 mL aqueous phase. Then, all mixtures were transferred to a dialysis bag (MWCO 8000–14,000 kDa; Solarbio, Beijing, China) and dialyzed at room temperature (RT) for 2 h to remove acetone and obtain the final UaNP solution. Coumarin-6 (C6)-labeled PCL-PEG nanoparticles (C6-UaNPs) were prepared using a similar method, except with 0.1 wt% C6 instead of UA. The UaNP and C6-NP solutions were then freeze-dried for 24 h (Beijing Sihuan Freeze Dryer, China) and stored at −20 °C until use. 

### 2.4. Preparation of RBCM Vesicles (RVs)

Fresh blood was collected from male BALB/c nude mice. To remove the plasma and yellow-brown buffy coat, blood was centrifuged at 300× *g* for 3 min at 4 °C (Nanjing Xianou Instrument Manufacturing Co., Ltd., Nanjing, China). Obtained RBCMs were washed with prechilled 1 × PBS 3 times (300× *g*, 3 min, 4 °C), then lysed for 30 min in an ice bath in hypotonic 0.25 PBS. The RBCMs were collected and subjected to repeated freeze–thaw cycles after being centrifuged at 8000× *g* for 15 min at 4 °C (Thermo Fisher Scientific, Waltham, MA, USA). Finally, the purified RBCMs were centrifuged at 8000× *g* for 8 min at 4 °C, suspended in water, and sonicated for 5 min in a capped glass vial using a bath sonicator. The supernatant contained the desired red blood cell membrane vesicles and was stored at −20 °C until use.

### 2.5. Preparation of UMNPs

After thawing the purified RV suspension at RT, the RV solution was centrifuged at 8000× *g* for 5 min at 4 °C to remove the supernatant. Then, RVs (1 mg/mL protein, 1 mL) were added to the UaNP solution (1 mg/mL, 1 mL) and sonicated for 20 min in an ice-water bath to coat the surface of UaNPs for the final UMNPs. RBCM-coated C6-UaNPs (C6-UMNPs) were prepared by a similar method except with 0.1 wt% C6 instead of UA.

### 2.6. Determination of Nanoparticle Size and Stability Analysis 

To investigate the particle size and zeta potential, the UaNP and UMNP solutions were first diluted with 1 mL ultrapure water, and the particle size, polymer dispersity index (PDI), and zeta potential of the samples were measured by dynamic light scattering (DLS; Bettersize, Shanghai, China). The morphology of UaNPs and UMNPs was observed by transmission electron microscope (TEM; JEOL JEM-1010, Tokyo, Japan). Briefly, 10 μL prepared UaNP and UMNP solutions were diluted with 20 μL ultrapure water and then dropped on a carbon-supported copper mesh. After 2 min, the excess water was removed using filter paper. After the samples were completely dried, they were observed by TEM.

Then, to investigate their stability behavior, UaNPs and UMNPs were re-dispersed in 1 × PBS (pH 6.5) and left at RT. Samples were taken out at 0, 2, 4, 6, 8, 12, 24, 36, and 48 h, and the particle size and PDI were measured by DLS to investigate their stability (*n* = 3).

### 2.7. Loading and Release of UA

The encapsulation efficiency (EE%) and drug-loading capacity (LC%) of UA nanoparticles were analyzed by high-performance liquid chromatography (HPLC; Shimadzu, Japan). Briefly, 10 mg UaNPs was dissolved in 600 μL acetone, then the solution was added to 600 μL deionized water. The mixture was centrifuged at 12,000 rpm for 15 min. The supernatant was collected and filtered with a 0.22 μm microporous membrane, and 20 μL filtrate was automatically injected into the HPLC system. The UA concentration was analyzed according to a predetermined calibration curve. The EE% and LC% of UA were calculated as follows:EE% = (mass of UA in NPs)/(total mass of UA in feed) × 100%(1)
LC% = (mass of UA in NPs)/(total mass of NPs) × 100%.(2)

To investigate UA release behavior, UaNPs (3 mg/mL, 1 mL) and UMNPs (10 mg/mL, 1 mL) were encapsulated in dialysis bags (MWCO, 8000 kDa) and placed in 20 mL PBS (pH 6.5) and PBS (pH 7.4), respectively, with shaking at 150 rpm for 48 h at 37 °C (Shanghai Zhicheng Analytical Instrument Manufacturing Co., Ltd., Shanghai, China). At 0, 0.5, 1, 1.5, 2, 3, 4, 6, 8, 12, 24, and 48 h, the released medium was collected, and the amount of released UA was measured by HPLC (*n* = 3).

### 2.8. Membrane Protein Verification of UMNPs

A 12% SDS-PAGE gel was prepared to verify the protein types and contents of RBCs, RBCMs, and UMNPs to determine whether UMNPs contained the protein content of RBCMs. The protein concentration of each sample was first determined using the BCA protein quantification kit, with RBCs, RBCMs, and UMNPs adjusted to the same concentrations. Then, 100 µL of each sample was combined with 25 µL of loading buffer. To denature the proteins, the mixtures were boiled for 5 min at 95 °C in a water bath. The prepared gel plate was then put into a different loading slot along with 15 µL of each group and 2 µL of protein marker. The gel pieces were stained with Coomassie brilliant blue for 30 min at RT following electrophoresis (Tanon Science and Technology, Shanghai, China), then they were rinsed with deionized water on the decolorizing shaker for 2 h (Kylin-Bell Lab Instruments, Nantong, China). 

### 2.9. Cellular Uptake Assay

A549 and RAW264.7 cells were cultured in 24-well plates (2 × 10^5^ cells/well). After culturing for 24 h, C6-NPs and RBCM-C6-NPs (20 μg/mL) were added to each plate and cultured for 24 h in the dark. Then, RVs and nuclei were labeled with DiO and DAPI, respectively (*n* = 3), and all samples were observed by confocal laser scanning microscopy (CLSM; ZEISS LSM710, Oberkochen, Germany). The excitation wavelengths of C6, DiO, and DAPI dyes were 523, 484, and 405 nm, respectively.

### 2.10. In Vitro Cytotoxicity Assay

A549 and NCI-H1975 cells were cultured in 96-well plates (5 × 10^3^ cells/well) for 24 h. Then, free UA, and UaNPs and UMNPs with various concentrations of UA (5, 10, 15, 20, 25, and 35 µg/mL) were added, and the cells were incubated for 24 and 48 h, respectively. Then, the cells were incubated with 0.5 mg/mL MTT for 4 h at 37 °C in the dark. The absorbance of samples was determined using a 96-well microplate reader at a wavelength of 490 nm (Tanon, Shanghai, China). Cell proliferation inhibition rates were calculated according to the following formulas:Proliferation inhibition ratio (%) = 1 − [(A_1_ − A_2_)/(A_3_ − A_2_)] × 100% (3)
where A_1_ is the average optical density (OD) value of the drug experimental group, A_2_ is average OD value of medium without cells, and A_3_ is average OD value of blank control group. 

GraphPad Prism software was used to perform the nonlinear regression analysis to calculate the 50% inhibitory concentration (IC50) value, which is the drug concentration that inhibits cell growth by 50% (*n* = 5).

### 2.11. Clone Formation Assay

A549 cells were cultured in six-well plates (200 cells/well). When cells were adherent, the solution of UA, UaNPs, and UMNPs (20 µg/mL UA in each formulation) was added to the wells. After incubation for 24 h, the liquid in the plates was taken out and replaced with fresh medium. When the cell density in solvent control reached >50 per cluster, cells were washed with PBS, fixed with 4% paraformaldehyde, and stained with 0.1% crystal violet. After washing with PBS again, cell colonies were visible to the naked eye. Then, cells counted from 5 randomly selected fields were visualized by microscopy (Nanjing Jiangnan Yongxin Optical Co., Ltd., Nanjing, China) at 100× magnification (*n* = 3). 

### 2.12. Cell Migration Assay

A549 cells were cultured in 12-well plates (5 × 10^5^ cells/well). When cells had grown into a confluent monolayer, the cell layer was scratched using a p200 pipette tip to initiate migration. Then, the cells were washed with PBS and incubated with UA, UaNPs, and UMNPs (20 µg/mL UA in each formulation) for 24 h and 48 h. Then, 4 fields (×20) were randomly chosen from each scratch wound and visualized by microscopy to evaluate the ability of cell migration (*n* = 3). 

### 2.13. Invasion Assay

For the cell invasion test, A549 cells at a density of 1.5 × 10^5^ suspended in serum-free medium with UA, UaNPs, and UMNPs (20 µg/mL UA in each formulation) were seeded into the upper chamber of a 24-well transwell membrane (PET membrane, diameter 6.5 mm, pore size 8 μm). Meanwhile, the lower chamber was co-cultured in a well with medium containing 10% FBS as an attractant. Cells were incubated at 37 °C and 5% CO_2_ as usual. After incubation for 48 h, the upper surface of the membrane was scrubbed to remove nonmigrating cells. A549 cells on the underside of the upper chamber were fixed with 4% paraformaldehyde for 30 min and stained with 0.1% crystal violet for 20 min, then washed with PBS 3 times. Multiple 10× magnification images per well were acquired, and the average counts were calculated (*n* = 3).

### 2.14. Apoptosis Assay

A549 cells in 6-well plates (1 × 10^6^ cells/well) were treated with UA, UaNPs, and UMNPs (20 µg/mL UA in each formulation) and incubated overnight. Subsequently, cells were harvested by trypsinization and then washed twice in ice-cold PBS. Then, the cells were resuspended in Annexin V Binding Buffer at a concentration of 1.0 × 10^7^ cells/mL, followed by 5 µL of APC Annexin V and 10 µL of propidium iodide solution. After gentle vortexing, cells were incubated in the dark for 15 min at RT. Apoptosis detection was performed with a FACSCalibur flow cytometer (BD Biosciences, Franklin Lakes, NJ, USA) and data were analyzed using FlowJo (FlowJo Studio, Carrboro, NC, USA).

### 2.15. Assessment of Reactive Oxygen Species (ROS) Production

A reactive oxygen species assay kit was used to assess the generation of intracellular ROS. After being treated with UA, UaNPs, and UMNPs (20 µg/mL UA in each formulation), A549 cells were washed with PBS 3 times. The cells were incubated with a working solution (1 µL reactive oxygen species assay kit solution diluted with 1000 µL DMSO) and then cultured at 37 °C with 5% CO_2_. After being cultured for 30 min, cells were washed again with PBS. Finally, the luminescence of fluorescent substances within the cells was photographed using CLSM to assess the amount of ROS.

### 2.16. Western Blot

A549 cells in 6-well plates (1 × 10^6^ cells/well) were treated with UA, UaNPs, and UMNPs (20 µg/mL UA in each formulation) and incubated. After incubating for 24 h, cells were treated with RIPA buffer, and total protein was determined by a BCA kit. A total of 30 μg of protein was loaded onto a 12% SDS-PAGE gel. After transfer, membranes were blocked by 5% TBST of non-fat milk and incubated overnight at 4 °C with rabbit anti-PARP, -P62, -LC3-I/II (CST, Kansas City, MO, USA), and mouse anti-GAPDH (Proteintech, Wuhan, China). The next day, the membranes were incubated with corresponding HRP-conjugated secondary antibodies. After washing, the membranes were finally covered with ECL chemiluminescent substrate (Tanon, China).

### 2.17. In Vivo Therapeutic Effect

Female BALB/c nude mice were subcutaneously injected with A549 cells (1 × 10^6^ cells/mouse). When the tumors reached approximately 100 mm^3^, the mice were randomized into four groups (*n* = 5) and intravenously injected with PBS and UA, UaNPs, and UMNPs with a UA concentration of 20 mg/kg. The tumor volume and body weight of mice were measured every 2 days. The tumor volume was calculated according to the following formula:Tumor volume = length × width^2^ × 0.52(4)

All mice underwent a 16-day treatment period with various medication formulations before being killed. Then, the tumors were taken and weighed, and major organs (heart, liver, spleen, lung, and kidney) were harvested, fixed with 10% formalin, embedded in paraffin, and labeled with hematoxylin and eosin (HE).

### 2.18. Immunohistochemical (IHC) Analysis

IHC labeling was used to identify Ki67- and P53-positive cells in tumor samples. Tris-EDTA buffer solution (pH 9.0) was used to pre-treat the tissue slices at 95 °C. A biotinylated immunoglobulin cocktail of goat anti-rabbit IgG was administered for 30 min at RT after samples had been incubated with the main antibody for 60 min. Visualization was carried out with the use of a polymer IHC detection device. Then, the percentage of Ki67- and P53-positive cells in each sample was calculated.

### 2.19. Statistical Analysis

All data are represented as mean ± standard deviation (SD). Student’s *t*-test was used to analyze differences between groups when comparing only two groups, with *p* < 0.05 indicating a significant difference.

## 3. Results

### 3.1. Synthesis and Characterization of UaNPs and UMNPs

After successful synthesis of UaNPs and UMNPs, DLS was used to measure their size and zeta potential. As shown in Figure 1a, Appendix A, the average size of UaNPs was 100.5 nm, which increased to 112.0 nm after camouflaging with RBCMs. Since the thickness of RBCMs was 11–12 nm, the increase in UMNP particle size confirmed the successful encapsulation of RVs. The average potential of UMNPs was −24.47 mV (Figure 1b), which was lower than that of UaNPs. This result also proved the successful encapsulation of RVs on the surface of UaNPs. TEM images show that UaNPs have a spherical shape (Figure 1c), while UMNPs show a core–shell morphology (Figure 1d). The core manifests as an area surrounded by a lighter gray shell showing RVs coated on the surface of nanoparticles. 

SDS-PAGE gel electrophoresis was used to verify whether the membrane proteins were altered during the process of preparing RVs and UMNPs. As shown in Figure 1e, the RV, RBCM, and UMNP groups presented the same protein bands and concentration, indicating that both RVs and UMNPs retained the natural RBCM proteins. 

Furthermore, the key protein of RBCs, especially the immunomodulatory CD47, can prevent the non-specific clearance of nanoparticles by MPS by encapsulating RVs on the nanodrug surface. Thus, the expression of CD47 on the surface of different nanoparticles was examined by Western blotting. It was observed that the UMNPs had an exterior protein environment that was similar to the RBCM source (Figure 1f, Appendix A), which prevented clearance by MPS, thereby prolonging the blood circulation of UA.

The stability data of UaNPs and UMNPs are given in Figure 1g. Both UaNPs and UMNPs showed excellent stability, with the particle size maintained at approximately 105.0 nm and 120.0 nm, respectively, within 24 h. Although the particle size and PDI increased slightly after 36 h, the results still indicate that UaNPs and UMNPs were stable in structure and distribution. HPLC analysis showed that nanoparticles had good encapsulation efficiency and drug loading capacity, at 62.22 ± 0.23% and 5.66 ± 0.07%, respectively. 

The drug-release property of UaNPs and UMNPs in PBS at pH 7.4 and 6.5 (simulating the slightly acidic tumor pH) is shown in Figure 1h. Approximately 58.4% and 60.7% of the UA was released from UaNPs and UMNPs in PBS at pH 6.5 over 48 h, higher than the amount released in PBS at pH 6.5. The prolonged and pH-sensitive release of UMNPs prevents UA degradation before targeting tumor tissues.

### 3.2. Cellular Uptake 

The effectiveness of drug delivery and targeting depends on how tumor cells or immune cells interact with nanoparticles. Thus, the in vitro cellular uptake process was observed using CLSM. In Figure 2a, the labeled nucleus, RVs, and nanoparticles are represented by fluorescence in blue, red, and green, respectively. Red and green fluorescence were seen to overlap the blue fluorescence after 24 h of administration, showing that the core–shell structure of UMNPs was preserved with tumor cells after being internalized by A549 cells. Comparing the quantitative uptake of UaNPs and UMNPs with RAW 264.7 cells using CLSM confirmed the immune evasion capacity of UMNPs. As shown in Figure 2b,c, a significant difference between UaNPs and UMNPs in RAW 264.7 cells could be seen, with mean fluorescence intensities of 68 and 49, respectively.

The poor cellular uptake of UMNPs by RAW 264.7 cells was significantly less than that of UaNPs (*p* < 0.01), mainly because of the proteins on the surface of RBCM-coated UMNPs. These proteins send a “don’t eat me” signal to the immune cell host to help UMNPs pretend to be autologous substances in order to avoid phagocytosis by macrophages. Hence, the RBCM coating avoided clearance by the immune system and improved the biocompatibility of UA, with adequate UA targeting of the tumor sites and exerting antitumor efficacy in NSCLC.

### 3.3. In Vitro Cytotoxicity, Migration, and Invasion Assays 

The growth-inhibiting effects of free UA, UaNPs, and UMNPs in A549 cells (Figure 3a,b) and NCI-H1975 cells (Figure 3c,d) were investigated by MTT assay. UMNPs inhibited the growth of A549 and NCI-H1975 cells in a dose- and time-dependent manner. UMNP treatment showed obvious inhibitory effects on the proliferation of A549 and NCI-H1975 cells. Furthermore, the evaluation of the A549 cell cloning assay showed that the group of UMNPs had far fewer colonies than free UA and UaNPs after 48 h incubation (Figure 3e,f). This indicates that UMNPs had superior antitumor activity compared to free UA and UaNPs.

A series of experiments was further carried out to investigate the anti-metastatic activity of UMNPs. To evaluate the impact of UMNPs on cell migration, the wound healing assay and transwell migration assay were carried out successively. It was found that UaNPs and UMNPs effectively suppressed the migration of A549 cells, presenting better antimetastatic activity than free UA (Figure 3g,h). Consistent with this finding, the transwell invasion assay showed that UMNPs significantly weakened the invasion capacity of A549 cells compared to free UA (Figure 3i,j). These results suggest that UMNPs greatly inhibit migration and invasion, which is important in inhibiting the tumor metastasis of NSCLC cells.

### 3.4. UMNPs Induce Apoptotic and Autophagic Cell Death in NSCLC Cells

Since the antitumor and antimetastatic effects of different UA formulations on NSCLC cells were observed, their potential mechanisms were further investigated. Based on the observation that free UA inhibited NSCLC cells by inducing apoptosis, whether UMNPs could induce apoptosis in cells was investigated by annexin V and PI double staining. The effect of UA, UaNPs, and UMNPs on the apoptosis of A549 cells was detected by flow cytometry. As shown in Figure 4a,b, treatment with UA, UaNPs, and UMNPs at 10 μg/mL for 48 h resulted in 17.1, 21.9, and 58.1% apoptotic cells, respectively, with baseline apoptosis of solvent control cells of 2.49% (*p* < 0.001). These results indicate that UMNPs could induce apoptosis better than UaNPs and free UA. Moreover, UMNPs activated PARP cleavage (89 kDa) in A549 cells (Figure 4c, Appendix A), which indicates that UMNPs trigger the caspase cascade by inhibiting the expression of PARP. We also examined whether UMNP-induced apoptosis was associated with intracellular ROS activation. Thus, intracellular ROS production in A549 cells after treatment with UMNPs for 48 h was measured. We found that treatment with UMNPs resulted in significantly higher intracellular ROS levels compared with PBS, UA, and UaNPs (Figure 4d), which is consistent with the induction of cell apoptosis analyzed by Western blot. Thus, intracellular ROS activation may potentially be involved in the induction of cell apoptosis by UMNP treatment. All results show that UMNPs significantly enhanced the UA-induced apoptosis of A549 cells in vitro.

Apart from apoptosis, autophagy also plays an important role in antitumor and anti-metastatic activities. UA has been reported to induce autophagy in NSCLC cells. Thus, to investigate whether UMNPs affect the dynamic process of autophagy in A549 cells, we performed Western blot to analyze the autophagic flux induced by UA, UaNPs, and UMNPs (Figure 4c, Appendix A). In contrast to the mock control, which did not induce LC3 lipidation (LC3II, a key molecular marker of autophagy) at 48 h, UMNPs significantly induced autophagy, as indicated by the increased expression of LC3II. The decreased level of P62 (an autophagy-related protein) further verified the autophagy by UMNPs. These results indicate that UMNPs have antitumor and antimetastatic activity by inducing apoptosis and autophagy in NSCLC cells.

### 3.5. In Vivo Antitumor Efficacy and Biosafety

Based on the excellent antitumor capacity in vitro, we further investigated antitumor efficacy in vivo in BALB/c nude mice bearing A549 cells with different treatments. As shown in Figure 5a, mice treated with PBS showed rapid tumor growth with a volume greater than 250 mm^3^ after a 16-day treatment period, while all UA formulations alleviated the rapid tumor growth. Treatment with UaNPs exhibited a much smaller tumor volume than free UA at the end of the treatment. Notably, slight tumor growth was observed in the group of UMNPs. Similarly, the appearance and weight of tumor sections at the end of the therapy period presented the same result (Figure 5b,c). UMNP treatment showed the best tumor inhibition efficacy, with a tumor inhibition rate (TIR) of 35% (Appendix A), which was much higher than that of UaNPs (26%) and free UA (20%). IHC analysis showed a low expression of Ki67 (a key protein related to tumor progression) in tumor tissues in the group treated with UMNPs compared with the control and UA groups (Figure 5e). In addition, UMNPs showed better P53 activating ability than the other treatments. All of the above results indicate that biomimetic UaNPs have excellent antitumor efficiency.

Furthermore, the biosafety of biomimetic RBCM-mediated UA treatment both in vitro and in vivo was investigated. As shown in Appendix A, the hemolytic activity of UA, UaNPs, and UMNPs was almost negligible. In addition, there was no significant weight change in all groups, indicating the good biocompatibility of UMNPs (Figure 5d). At the end of treatment, all mice were sacrificed, and their major organs (heart, liver, spleen, lung, kidney) were taken for hematoxylin and eosin (HE) analysis. As shown in Figure 5f, organs treated with UMNPs presented negligible damage, further verifying the favorable biocompatibility of UMNPs.

## 4. Discussion

According to the latest global cancer statistics, lung cancer, which is among the cancers with the highest incidence, still has the highest mortality globally [1]. As the most common lung cancer, NSCLC is known to have poor clinical outcomes [34]. Various advancements in NSCLC therapy have been developed in recent decades. Traditional Chinese medicine has unique advantages to provide more chances for developing NSCLC treatment [35]. However, the poor water solubility, systemic toxicity, and other features limit the clinical application of several traditional Chinese medicines [36,37]. In our previous study, we developed multiple nanosystems to deliver traditional Chinese medicine, including curcumin, tetrandrine, and paclitaxel, to enhance their solubility, improve targeting efficiency, reduce toxicity, and improve tumor-inhibiting effects [38,39,40]. Recently, UA has been reported as a potential anticancer candidate, but is not clinically applied due to its poor solubility, unsatisfactory bioavailability, and non-selective toxicity [14].

In this study, we designed a novel biomimetic nanoplatform for delivering UA by loading UA into PEG-PCL nanoparticles on which bionic RBCMs were encapsulated on the surface. First, PEG-PCL nanoparticle-loaded UAs (UaNPs) were synthesized by the nanoprecipitation method in a uniform, regular spherical shape under TEM observation. Then, RBCMs taken from mice were prepared as RVs to coat the surface of UaNPs to obtain the final biomimetic UMNPs. UMNPs presented a core–shell structure, indicating the presence of RBCMs on the surface. A similar result was found in the analysis of the particle size and zeta potential of UaNPs and UMNPs, showing values of 100.5 and 112.0 nm, and −17.23 and −24.47 mV, respectively, which remained stable over 48 h. The EE% and LC% of nanoparticles were 62.22 ± 0.23% and 5.66 ± 0.07%, respectively. The release behavior of UaNPs and UMNPs showed that both achieved sustained release in a medium mimicking natural physiology and the tumor environment. 

The proteins of RBCMs are important in avoiding the clearance of the immune system and prolonging the blood circulation of drugs. Thus, proteins on the surface of UMNPs were analyzed by SDS-PAGE gel electrophoresis. The results show that UMNPs retain similar protein species and concentrations to RBCs and RBCMs. In addition, the expression of CD47, which sends “don’t eat me” signals to the immune system, was further investigated by Western blot. There was no significant difference between RBCMs and UMNPs, indicating that RBCM-coated UMNPs have immune-evading capability.

As expected, RAW 264.7 cells demonstrated significantly less intracellular uptake of UMNPs than UaNPs, as visualized by CLSM. It was further confirmed that UMNPs presented more cellular uptake than UaNPs. The encapsulation of RBCMs on the surface of UaNPs may help them escape the non-special clearance of MPS, and then prolong the UA blood circulation and target tumor sites to finally exert tumor-inhibiting activity.

In vitro antitumor analysis was then carried out by different methods. As investigated by MTT, all UA formulations were found to inhibit the growth of A549 and NCI-H1975 cells in a dose- and time-dependent manner. The tumor cell inhibition of UMNPs was not ideal compared with the free UA in A549 and NCI-H1975 cells after 24 h treatment, but UMNPs showed a better tumor cell inhibiting effect than free UA in NCI-H1975 cells after 48 h treatment. Furthermore, UMNPs exhibited the highest antitumor efficacy among the three UA formulations, according to cloning formation assay. Then, migration and transwell invasion assay proved that UMNPs not only inhibited NSCLC cells but also showed the best antimetastatic activity compared to other formulations. 

Considering the fact that apoptosis and autophagy are the main anticancer mechanisms, and multiple studies have demonstrated that UA can induce these processes, we examined whether different UA nanoparticles could kill cancer cells by inducing apoptosis or autophagy. Exposure to UMNPs induced much more apoptosis and ROS generation than free UA and UaNPs to trigger cancer cell death. The autophagy assay results suggest that UaNPs enhance antitumor activity through the activation of the autophagic pathway.

An A549-bearing BALB/c nude mouse model was employed to investigate the antitumor activity of three UA formulations. The photos of tumors, tumor growth curves, tumor weight, and IHC analysis at the end of 16-day therapy indicated that UMNPs displayed the greatest inhibition ability against NSCLC. Although the excellent cytotoxicity of UA nanoparticles was observed, the safety and biocompatibility of UA are of great concern regarding its application in the clinic. Results of body weight changes and the HE assay of major organs suggest that UMNPs and UaNPs do not cause severe systemic toxicity. 

The biomimetic UMNP nanoplatform offers a novel drug-delivery method for UA and broadens the range of therapeutic strategies for NSCLC, but further in vivo evaluation should be carried out to ensure the therapeutic effect and safety of these agents.

## 5. Conclusions

In summary, by coating with RBCMs, we developed a novel biomimetic drug-delivery platform loading UA with better stability, biosafety, and tumor-targeting. With the coating, UMNPs reduced non-specific immune clearance by the mononuclear phagocyte system (MPS) and realized prolonged long-term blood circulation because of the CD47 on the RBCM surface. Furthermore, RBCM-coated UA nanoparticles with smaller sizes could target tumor tissue owing to the enhanced permeability and retention (EPR) effect. Compared to free UA, biomimetic membrane-based nanoparticles (UMNPs) exhibited outstanding antitumor efficacy both in vitro and in vivo. We further found that UMNPs improved antitumor activity in vitro by inducing the apoptosis and autophagy of A549 cells. In addition, RBCM-coated UMNPs presented superior biocompatibility in mice. Although more analysis should be carried out to explore the antitumor mechanism of UMNPs in vivo, our study was the first to coat RBCM on the surface of UA nanoparticles successfully. This new kind of biomimetic drug-delivery system improves the superior antitumor efficiency of NSCLC models and provides a promising strategy for delivering UA to improve the treatment of NSCLC in the future.

## Figures and Tables

**Figure 1 cancers-14-04520-f001:**
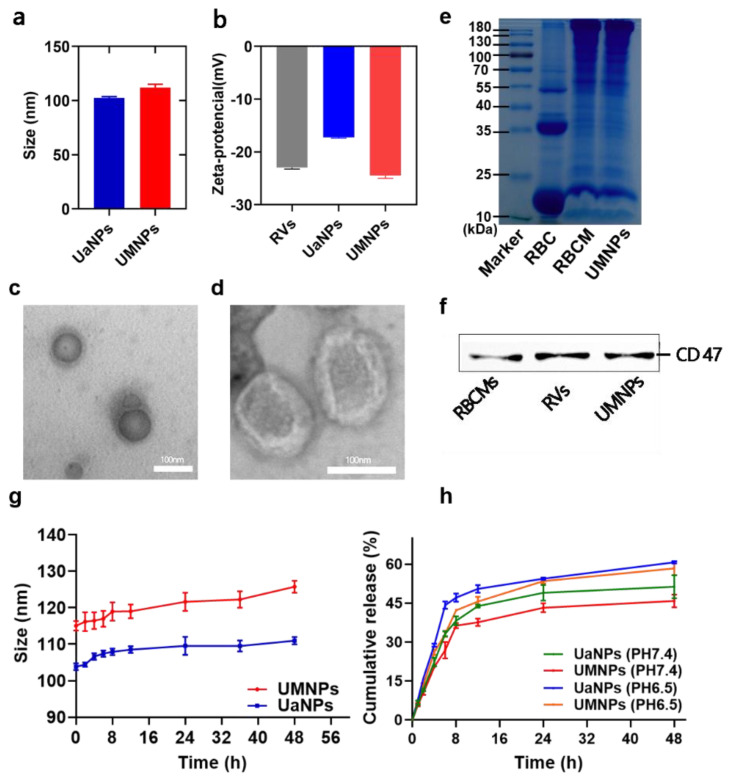
Characterization of NDDS. (**a**,**b**) Particle size and zeta potential of UaNPs and UMNPs, respectively; (**c**,**d**) TEM images of UaNPs and UMNPs, respectively. Scale bar = 100 nm; (**e**) Total protein verification in RBC, RBCMs, and UMNPs by SDS-PAGE; (**f**) Western blot analysis of RBCMs and UMNPs for characteristic RBCM marker CD47; (**g**) Particle size changes of UMNPs in 10% FBS within 48 h; and (**h**) Release curve of UaNPs and UMNPs in vitro in PBS (pH 7.4 and 6.5). Data presented as mean ± SD (*n* = 3).

**Figure 2 cancers-14-04520-f002:**
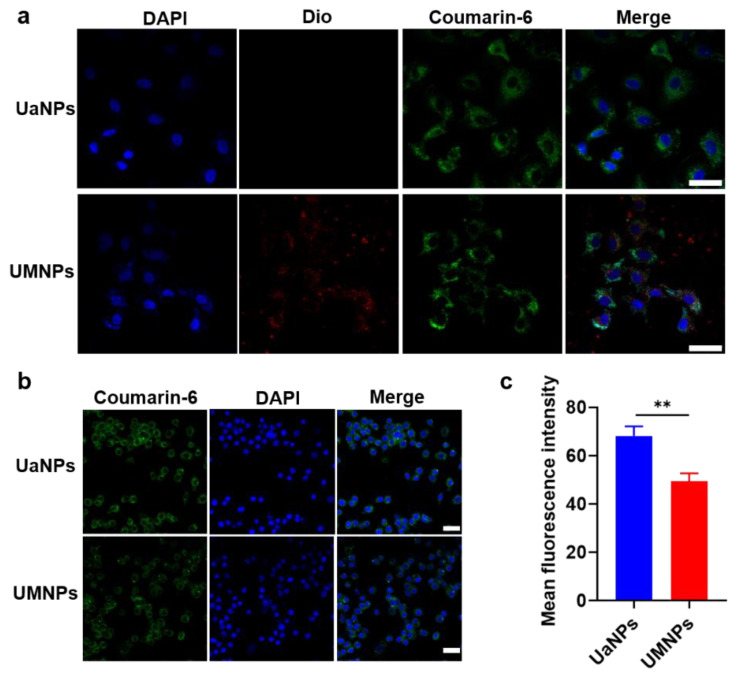
Confocal images of cellular uptake of UaNPs and UMNPs with (**a**) A549 cells and (**b**) RAW264.7 cells. Scale bar = 20 μm. (**c**) Quantitative analysis of fluorescence intensity. Scale bar = 20 μm. Differences were considered significant when ** *p* < 0.01.

**Figure 3 cancers-14-04520-f003:**
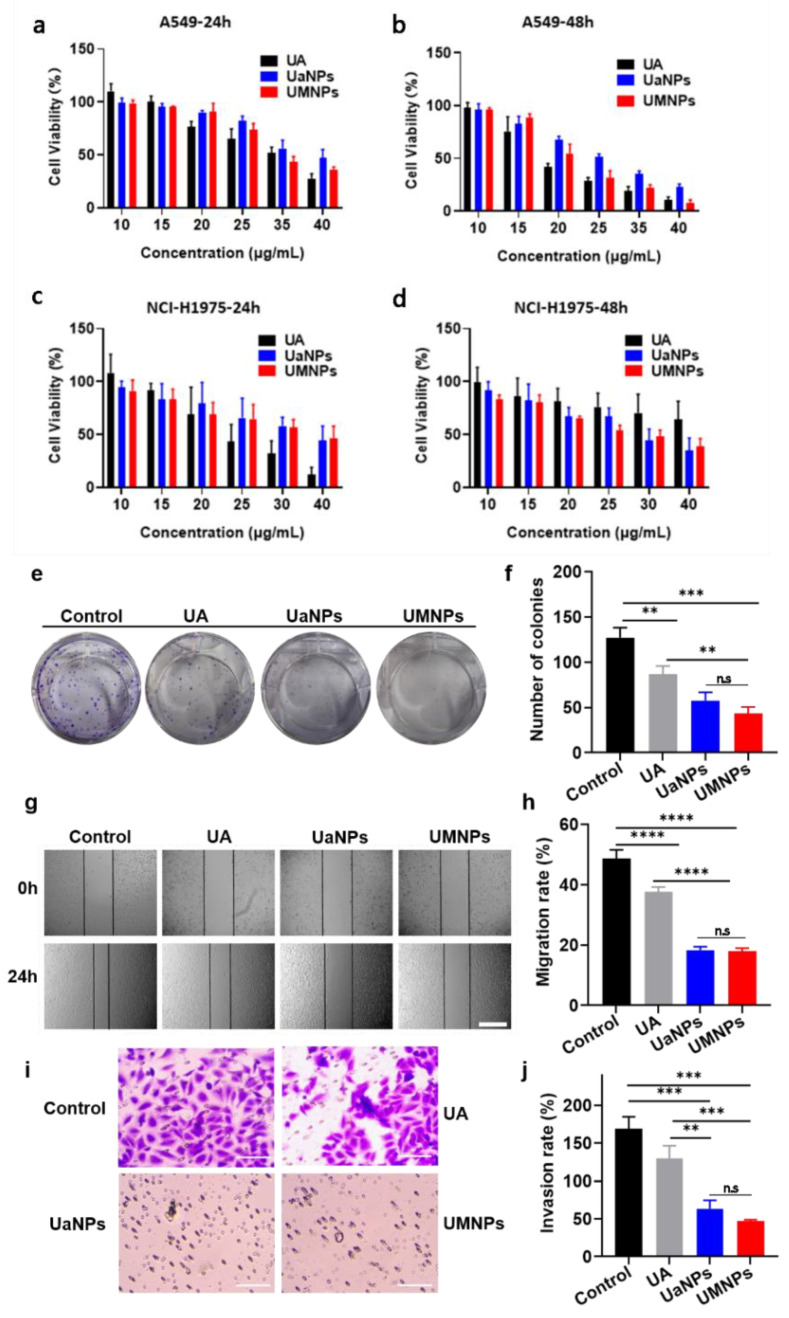
In vitro cytotoxicity of UA, UaNPs, and UMNPs: (**a**,**b**) Cell survival of A549 cells after treatment with different concentrations of free UA, UaNPs, and UMNPs for 24 h and 48 h, respectively; (**c**,**d**) Cell survival of NCI-H1975 cells after treatment with different concentrations of free UA, UaNPs, and UMNPs for 24 h and 48 h, respectively; (**e**,**f**) Cloning formation assay of free UA, UaNPs, and UMNPs in A549 cells; (**g**,**h**) Cell migration assay of free UA, UaNPs, and UMNPs in A549 cells. Scale bar = 200 μm; (**i**,**j**) Transwell invasion assay of free UA, UaNPs, and UMNPs in A549 cells. Scale bar = 100 μm. Differences were considered significant when ** *p* < 0.01, or *** *p* < 0.001, **** *p* < 0.0001, Respectively, while there is no significant when *p* > 0.05 which means ns.

**Figure 4 cancers-14-04520-f004:**
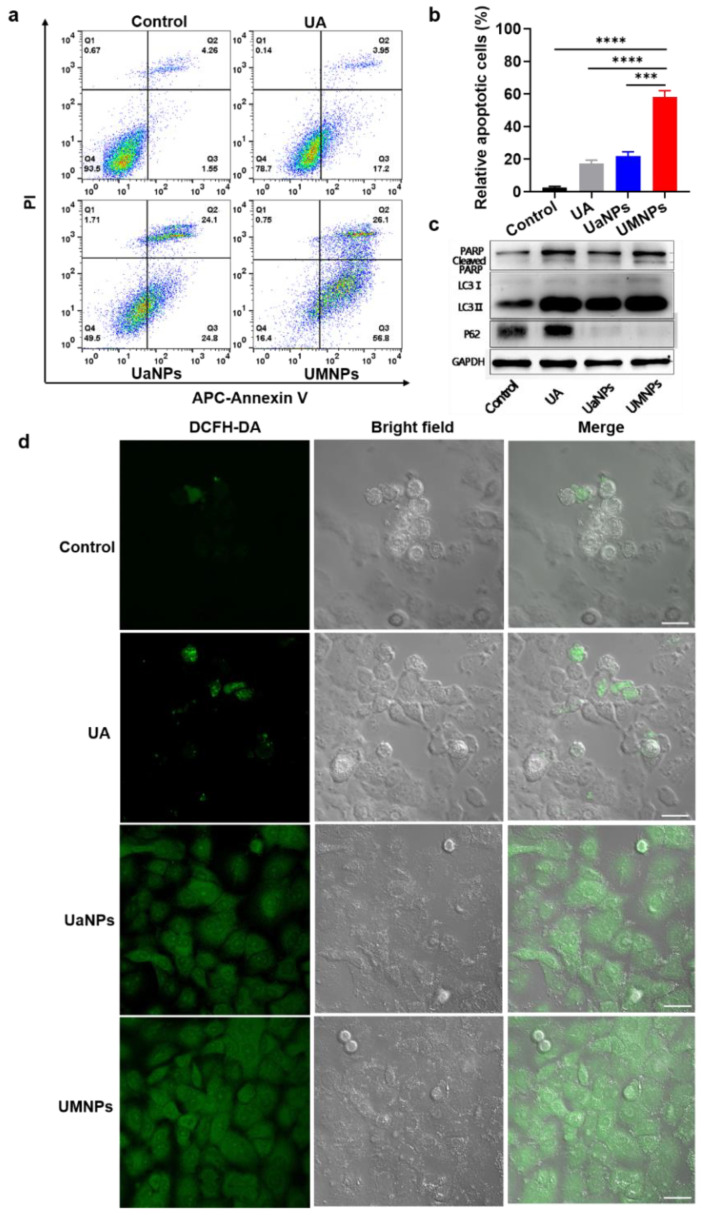
UA, UaNPs, and UMNPs induce apoptotic and autophagic cell death in NSCLC cells. (**a**,**b**) Flow cytometry analysis to detect the apoptotic effect of free UA, UaNPs, and UMNPs in A549 cells for 48 h. (**c**) Western blotting was used to detect expression of apoptosis- and autophagy-related proteins in A549 cells treated with UA, UaNPs, and UMNPs for 48 h. (**d**) Alteration of intracellular reactive oxygen species levels by UMNP treatment. Control for A549 cells with DCFH-DA treatment. Scale bar = 20 μm. Differences were considered significant when *** *p* < 0.001 or **** *p* < 0.0001, respectively.

**Figure 5 cancers-14-04520-f005:**
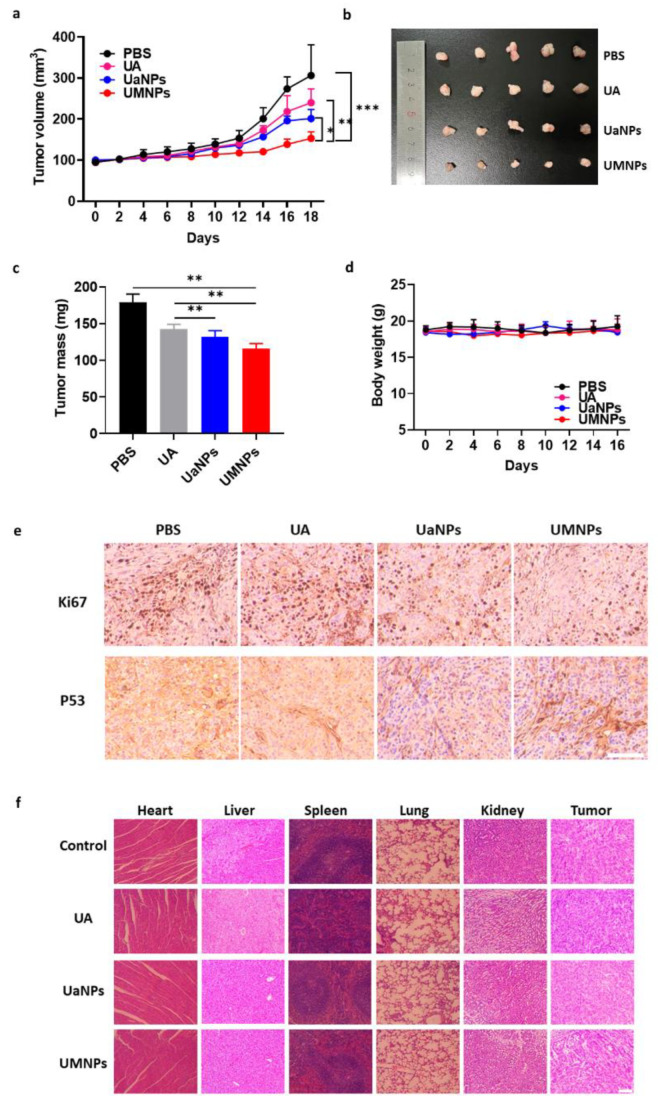
In vivo antitumor efficacy and biosafety of UA formulations. (**a**) Tumor volume of A549-bearing mice treated with different UA formulations over 16 days; (**b**) Images of tumor sections at the end of therapy period; (**c**) Weight of tumor sections in different groups at end of therapy period; (**d**) Body weight of A549-bearing mice treated with PBS, UA, UaNPs, and UMNPs; (**e**) IHC analysis of tumor tissues after treatment with different UA formulations. Scale bar = 200 μm; and (**f**) HE staining of heart, liver, spleen, lung, and kidney of mice treated with different UA formulations. Scale bar = 200 μm. Data presented as mean ± SD (*n* = 5). Differences were considered significant when * *p* < 0.05, ** *p* < 0.01, or *** *p* < 0.001, respectively.

## Data Availability

All data generated or analyzed during this study are included in this manuscript. Further inquiries can be directed to the corresponding authors.

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
