# Peer review of "Biomimetic Red Blood Cell Membrane-Mediated Nanodrugs Loading Ursolic Acid for Targeting NSCLC Therapy"

_cancers, 2022, doi:10.3390/cancers14184520_

Round 1

Reviewer 1 Report

Wu et al describe in this manuscript on how they develop nanodrug of encapsulated Ursolic acid coated with biodegradable PEG-PCL and red blood cell membrane for NSCLC treatment. Though this topic is of interest, the language quality in which this manuscript was written needs robust editing. On a positive note, data quality are good and experiments were well-designed. In its current form, the manuscript is not suitable for publication.

Below are comments for the authors (unfortunately, there are too many language errors to list here)

Line 130 – A549 and H1975 are human cells, not murine.

Line 87 – Instead of using “sends out a”, a better word would be “represents” or “serves as”

Line 103 – “NSCLC” and not “NSCLS”

Line 138 – “dissolves” and not “solves”

Line 160 – please spell out what “RVs” mean.

Line 167 – A more accurate title of 2.6 should be “Determination of Nanoparticle Size and Stability Analysis”

Line 222 – 2.10 should be “In vitro Cytotoxicity Assay”

Line 242 – “each well respectively” should be replaced with “respective wells”

Line 245 - “cloning of the cell was” to be replaced with “cell colonies were”

Line 265 – “The upper chamber was…” should be replaced with “…Cells on the underside of the upper chamber were…”

Line 287 – What does CLSM stand for?

Line 297 – “treated” should be replaced with “covered”

Line 341 – “hurt” should be replaced with “altered”

Line 352 – What is PDI?

Line 390 – 3.3 should be “In vitro cytotoxicity, migration and invasion assays”

Fig 3a-d - Can the authors provide an explanation why cell viability of UaNPs and UMNPs are similar in H1975 but not in A549 cells?

The authors should provide the IC50 values of UA, UaNPs and UMNPs for A549 and H1975 cells.

The authors should provide a statement or two on why Ki67 and p53 were used in IHC analysis in Results.

The authors should indicate in Legends what *, **, *** and **** represent in the figures. If there is no significant difference, please put in “n.s” (not significant) in all figures.

Can the authors adjust the image quality for Fig 3g? The cells and lines cannot be seen.

How were the mice sacrificed?

Can the authors provide the calculation on how they determined 5 mice per group?

The title of manuscript can be changed into something like – “Red cell membrane-coated Ursolic Acid-PEG-PCL as Biomimetic Nanodrug for NSCLC therapy”?

Author Response

Dear Ms. Lora Zheng and reviewer,        

Thanks for providing us with this great opportunity to submit a revised revision of our manuscript (Manuscript ID: cancers-1855416). We appreciate the detailed and constructive comments provided by the reviewer. We have carefully considered the suggestion of the reviewer and tried our best to improve the manuscript. 

Responds to the reviewers' comments:

Reviewer #1 l 

Comment 1: The language quality in which this manuscript was written needs robust editing.

Response: We have revised the English of the manuscript according to the reviewer. Besides, the “RV” means RBCMs-Vesicles (RVs) which is in line 146, “CLSM” means confocal laser scanning microscopy which is in line 217, and “PDI” means polymer dispersity index which is in line 168.

Comment 2: Fig 3a-d - Can the authors provide an explanation why cell viability of UaNPs and UMNPs are similar in H1975 but not in A549 cells?

Response: As suggested by the reviewer, the reason may be the unstable state of the cells at that time or an operational error during the fabrication of UMNPs. Therefore, we re-assessed the toxicity of UA, UaNPs, and UMNPs for A549 cells and the results are as follows:

Comment 3: The authors should provide the IC50 values of UA, UaNPs and UMNPs for A549 and H1975 cells.

Response: The IC50 values of UA, UaNPs, and UMNPs for A549 and H1975 cells were shown as follows:

The IC50 values of different formulations for A549

UA

UaNPs

UMNPs

24h

25.09 μg/mL-1

31.51 μg/mL-1

26.50 μg/mL-1

48h

14.23 μg/mL-1

20.13 μg/mL-1

16.37 μg/mL-1

The IC50 values of different formulations for H1975

UA

UaNPs

UMNPs

24h

27.14 μg/mL-1

35.46 μg/mL-1

25.79 μg/mL-1

48h

25.33 μg/mL-1

27.46 μg/mL-1

25.94 μg/mL-1

Comment 4: The authors should indicate in Legends what *, **, *** and **** represent in the figures. If there is no significant difference, please put in “n.s” (not significant) in all figures.

Response: The legends of figures have been revised according to the suggestion.

Comment 5: Can the authors adjust the image quality for Fig 3g? The cells and lines cannot be seen.

Response: The brightness, contrast, and exposure of Fig 3g were adjusted in the manuscript.

Comment 6: How were the mice sacrificed?

Response: All mice were sacrificed by the nedeck thrust dislocation method.

Comment 7: Can the authors provide the calculation on how they determined 5 mice per group?

Response: Referring to other articles, most of them have 5 mice per group in vivo antitumor efficacy evaluation, so we also set n=5 in each group[1-3].

Comment 8: The title of manuscript can be changed into something like – “Red cell membrane-coated Ursolic Acid-PEG-PCL as Biomimetic Nanodrug for NSCLC therapy”?

Response: We are very grateful to the reviewer for giving us such pertinent advice. According to our deliberate consideration, the title “Biomimetic Red Blood Cell Membranes-mediated Nanodrugs Loading Ursolic Acid for Targeting NSCLC Therapy” can be changed into “Red cell membrane-coated Ursolic Acid-PEG-PCL as Biomimetic Nanodrug for NSCLC therapy”.

We hope that the revised version of the manuscript is now acceptable for the journal of Cancers. Please also advise if the manuscript needs any additional information.

Thank you and best regards,

Sincerely,

Dr. Huae Xu

Affiliated Cancer Hospital of Nanjing Medical University, Jiangsu Cancer Hospital &

Department of Pharmaceutics, School of Pharmacy, Nanjing Medical University,

Nanjing, China

Email: xuhuae@njmu.edu.cn

Reference:

  1. Wang, Y.Q.; Huang, C.; Ye, P.J.; Long, J.R.; Xu, C.H.; Liu, Y.; Ling, X.L.; Lv, S.Y.; He, D.X.; Wei, H.; et al. Prolonged blood circulation outperforms active targeting for nanocarriers-mediated enhanced hepatocellular carcinoma therapy in vivo. J Control Release 2022, 347, 400-413, doi:10.1016/j.jconrel.2022.05.024.
  2. Shen, Z.; Song, J.; Yung, B.C.; Zhou, Z.; Wu, A.; Chen, X. Emerging Strategies of Cancer Therapy Based on Ferroptosis. Adv Mater 2018, 30, e1704007, doi:10.1002/adma.201704007.
  3. Zhang, Y.; Xia, Q.; Wu, T.; He, Z.; Li, Y.; Li, Z.; Hou, X.; He, Y.; Ruan, S.; Wang, Z.; et al. A novel multi-functionalized multicellular nanodelivery system for non-small cell lung cancer photochemotherapy. J Nanobiotechnology 2021, 19, 245, doi:10.1186/s12951-021-00977-3.

Reviewer 2 Report

In this manuscript, the authors developed a biomimetic drug delivery system based on the UA-loaded nanoparticles (UaNPs) with Red Blood Cell Membrane (RBCM) coating for better stability, biosafety, and tumor targeting. The authors showed that the RBCM coating reduced nonspecific immune clearance by the mononuclear phagocyte system and led to long-term circulation in blood. The authors suggested that biomimetic membrane-based nanoparticles UMNPs exhibited outstanding anti-tumor efficiency and superior biocompatibility in mice. Based on data from body weight, hematoxylin and eosin (HE) analysis, and IHC analysis, the authors concluded that UMNPs were safe with no toxicity.

This manuscript is poorly written, with many unclear sentences and grammatical errors. Serious work is needed to improve the writing. Additionally, there are significant experimental issues that need to be addressed. For example, the quality of Western data is poor. Quantitative data should be presented for IHC in Fig. 5e. Functional data for liver and blood should be shown in order to conclude that UA and particles do not exhibit toxicity in mice. Also, there is no data showing that the particles actually persisted in blood for a long period, which authors stated in Conclusions.

Author Response

Dear Ms. Lora Zheng and reviewer,

Thanks for providing us with this great opportunity to submit a revised revision of our manuscript (Manuscript ID: cancers-1855416). We appreciate the detailed and constructive comments provided by the reviewer. We have carefully considered the suggestion of the reviewer and tried our best to improve the manuscript. 

Responds to the reviewers' comments:Reviewer #2 

  • Comment 1: This manuscript is poorly written, with many unclear sentences and grammatical errors. Serious work is needed to improve the writing.

Response: Thank you for the detailed review. We have carefully and thoroughly proofread the manuscript to correct all the grammar and typos.

  • Comment 2: Additionally, there are significant experimental issues that need to be addressed. For example, the quality of Western data is poor.

Response: Thanks for your great suggestion on improving the accessibility of our manuscript. The Western blotting was further refined.

  • Comment 3: Quantitative data should be presented for IHC in Fig. 5e.

Response: Thanks for your great suggestion. The quantitative data for IHC was completed as follows.

  • Comment 4: Functional data for liver and blood should be shown in order to conclude that UA and particles do not exhibit toxicity in mice. Also, there is no data showing that the particles actually persisted in blood for a long period, which authors stated in Conclusions.

Response: We agree with you that the toxicity of UaNPs and UMNPs is important for its use. However, PEG-PCL was a biodegradable drug delivery system and was approved by Food and Drug Administration (FDA) for delivering drugs in various biomedical applications [1,2]. The RBCMs were collected from mice that are non-toxic. Furthermore, the mice showed no significant toxicity in terms of hair glossiness, mental changes, and body weight change during the treatment. Since we didn't have enough time to complete the in vivo biosafety evaluation which requires at least 21 days of dosing. We assessed the hemolytic activity of different UA formulations in vitro.

We hope that the revised version of the manuscript is now acceptable for the journal of Cancers. Please also advise if the manuscript needs any additional information.

Thank you and best regards,

Sincerely,

Dr. Huae Xu

Affiliated Cancer Hospital of Nanjing Medical University, Jiangsu Cancer Hospital &

Department of Pharmaceutics, School of Pharmacy, Nanjing Medical University,

Nanjing, China

Email: xuhuae@njmu.edu.cn

Reference:

  1. Grossen, P.; Witzigmann, D.; Sieber, S.; Huwyler, J. PEG-PCL-based nanomedicines: A biodegradable drug delivery system and its application. Journal of Controlled Release 2017, 260, 46-60, doi:https://doi.org/10.1016/j.jconrel.2017.05.028.
  2. Danhier, F.; Ansorena, E.; Silva, J.M.; Coco, R.; Le Breton, A.; Préat, V. PLGA-based nanoparticles: An overview of biomedical applications. Journal of Controlled Release 2012, 161, 505-522, doi:https://doi.org/10.1016/j.jconrel.2012.01.043.

Round 2

Reviewer 1 Report

Though changes have been made, extensive revision of English is required to the revised manuscript. 

Reviewer 2 Report

No comments besides the previously submitted.